# The Effect of the Motivational Climate on Satisfaction with Physical Education in Secondary School Education: Mediation of Teacher Strategies in Maintaining Discipline

**DOI:** 10.3390/bs13020178

**Published:** 2023-02-15

**Authors:** Clara Marina Bracho-Amador, Antonio Granero-Gallegos, Antonio Baena-Extremera, Ginés David López-García

**Affiliations:** 1Department of Education, University of Almeria, 04120 Almeria, Spain; 2Health Research Centre, University of Almeria, 04120 Almeria, Spain; 3Department of Musical, Plastic and Corporal Expression, Faculty of Education Sciences, University of Granada, 18071 Granada, Spain

**Keywords:** discipline, motivational climates, physical education, satisfaction

## Abstract

The objective of this study was to analyze the mediating role of strategies to maintain classroom discipline between the motivational climate generated by the teacher and the students’ satisfaction with physical education classes. The research design was observational, descriptive, cross-sectional, and non-randomized. In total, 2147 secondary school physical education students participated (*M*_age_ = 15.05; *SD* = 1.45) (male = 1050; female = 1097). A structural equation model was calculated with latent variables controlled by the teacher’s sex and time of service and using the scales of the motivational climate, the teacher’s strategies for maintaining classroom discipline, and the students’ satisfaction with physical education classes. The results from the model highlight the importance of intrinsic strategies in maintaining discipline; these act as a mediator between the motivational climate towards learning and the students’ satisfaction with physical education classes. In addition, the findings reveal the influence of a performance-oriented climate in predicting boredom in a class when the teacher shows an indifference towards maintaining discipline.

## 1. Introduction

During adolescence, disruptive behaviors in the classroom pose a serious problem for the teaching/learning process [1] and can have an impact on satisfaction with school itself, with the negative impact on students that this entails at the academic, social, and emotional levels [2,3]. In the field of physical education (PE), studies trying to understand the mechanisms associated with disruptive behaviors within the classroom have been conducted, e.g., [2]. Specifically, the use of disciplinary behaviors by the teacher has been linked to a decrease in disruptive behaviors in PE classes [4]. For this reason, discipline has been considered one of the most important and difficult educational aspects to manage in subjects such as PE [4]. The teacher plays an important role in managing discipline in the classroom [5]; hence, correct discipline management in the classroom supposes positive attitudes, as well as greater enjoyment, satisfaction and learning of PE [6,7].

The satisfaction of students with the PE classroom can be important for creating interest in practicing sports, which is essential for students to acquire physical exercise habits [6]. In fact, it is thought that perceived satisfaction with PE classes might, to a greater extent, be influenced by the role that the teacher plays within the classroom. Some studies [8,9] show that the teacher’s role can be seen as one of the reasons for generating discipline in the PE classroom; so much so that some authors highlight the motivational climate generated by the teacher in the classroom as being responsible for the students’ successful or failing behaviors [10]. Previous surveys [8,11] have demonstrated the need to investigate in more detail the role played by the teacher in sustaining discipline and influencing the social environment in the classroom. However, to date, we are unaware of any research that has examined the effects of the PE-teacher-generated social climate in the classroom on subject satisfaction. Therefore, it seems important to analyze the potential mediating role that might be played by strategies to maintain discipline in class when it comes to inducing student satisfaction or boredom, taking into account the perception of a certain classroom climate.

### 1.1. Learning and Performance Orientations in PE Lessons

Achievement goal theory (AGT) [12] is one of the theories that can provide answers regarding the learning orientations present in the PE classroom. According to AGT, the individual is perceived as an organism directed by rational objectives that lead to a goal, and by demonstrating competence and ability in achievement contexts. In this case, a person’s achievement goals constitute the main mechanism for judging competency and determining their perception of achieving success. In PE, the perception of success or failure depends on recognizing the performance objective, mainly by distinguishing two motivational climates: the task-oriented climate (TC) and the ego-oriented climate (EC) [13,14]. TC refers to that centered on task-focused cooperation between learners, occurring when the situational characteristics of the teacher or classmates are mindful of supporting effort, and success is focused on both the learning process and the intrapersonal criteria related to effort and personal improvement. In contrast, EC is defined by mastery of the task at any cost, where success is based on normative and interpersonal criteria, with a punitive response to errors and rivalry among students [12]. Students who perceive a TC in PE as an end in itself prefer challenging tasks. They tend to have fun in class, be more satisfied with the subject [15] and perceive intrinsic reasons for maintaining discipline [16]. Conversely, an EC will create an environment in which the students try to gain social approval within the class group [17]. As a result, the students will be less satisfied with the subject and perceive introjected or indifferent reasons for maintaining discipline [16]. In this sense, we consider the possible role of the motivational classroom climate as a proximal factor in the strategies of PE teachers to maintain class discipline [18,19].

### 1.2. Strategies to Maintain Discipline

Strategies for maintaining classroom discipline have been part of the pedagogical approaches to PE lessons [20,21]. According to the outcomes of the cognitive evaluation theory (CET), strategies for maintaining discipline can be located on a self-determination continuum, which includes the following reasons [22,23]: (i) intrinsic reasons, which produce pleasure, satisfaction or excitement while the person performs an activity; (ii) introjected reasons, determined by internal pressures aimed at avoiding negative assessment, guilt or shame; (iii) the teacher’s indifference towards maintaining discipline and towards the prevailing reasons for the students feeling a lack of control. In this regard, authors Papaioannou [23] and Gutiérrez and López [24] examined the relationship between the perceived motivational climate and the perceived strategies by teachers for maintaining discipline, arguing that a TC was positively related to perceived teaching strategies that promote the most self-determined reasons (i.e., intrinsic reasons) for discipline. In contrast, EC mobilizes strategies that promote introjected reasons, as well as perceptions of indifference towards maintaining discipline. Likewise, Gutiérrez and López [24] stated that the reasons for being disciplined in the classroom can vary from intrinsic reasons to an indifference towards maintaining discipline. Students who perceive more self-determined reasons from the teacher will have greater satisfaction with PE classes, while students who perceive less self-determined reasons will become bored in PE classes.

### 1.3. Satisfaction with Physical Education

Current scientific evidence emphasizes the importance of ensuring students feel satisfied with PE classes [6,21,25]. Students’ satisfaction with classes can influence their stress [26], social relationships [27], and academic performance [28,29]. Conversely, dissatisfaction has been associated with negative behaviors such as anxiety [30], depression [31], and even dropout behaviors [32]. Despite its importance, we have failed to find any studies analyzing the predictive effects of motivational climates (i.e., EC and TC) on satisfaction with PE that also consider the possible role of the perception of the discipline strategies used by the teacher in the classroom.

### 1.4. The Present Study

Based on all of the above, the objective of this study was to analyze the mediating role of strategies to maintain classroom discipline between the motivational climate generated by the teacher and the students’ satisfaction with PE classes. The following hypotheses were established: First, TC is positively related with satisfaction and negatively related with boredom with PE classes (H1); second, that EC is positively related with boredom and negatively related with satisfaction with PE classes (H2); third, the intrinsic reasons for maintaining discipline act as a positive mediator between TC and students’ satisfaction with PE classes (H3); fourth, that indifference and introjected reasons for maintaining discipline act as a negative mediator between TC and satisfaction with PE classes (H4); fifth, that indifference and introjected reasons for maintaining discipline act as a positive mediator between EC and boredom with PE classes (H5); sixth, that the intrinsic reasons for maintaining discipline act as a negative mediator between EC and satisfaction with PE classes (H6) (Figure 1).

## 2. Materials and Methods

### 2.1. Design and Participants

The research design was observational, descriptive, cross-sectional and non-randomized. An a priori analysis of the statistical power of the adequate sample size for meeting the study’s objective was carried out using the Free Statistics Calculator v.4.0 [33]. It was estimated that a minimum of 2083 participants were needed for the f^2^ = 0.128 effect sizes (ES) with a statistical power of 0.99 and a significance level of α = 0.05 in a structural equation model with seven latent variables and 59 observable variables. The following inclusion criteria were established to participate in this study: (i) to be a student of the second, third, or fourth grade of compulsory secondary education (CSE), or first year of high school (baccalaureate) of a public secondary school from the autonomous region of Andalusia; (ii) to provide informed consent for participation; (iii) regular attendance at classes; and (iv) fill out the questionnaire completely. A total of 2147 secondary school students (male = 1050; female = 1097) from 18 public secondary schools located in urban and rural contexts from the autonomous region of Andalusia, in the provinces of Almería (17.6%), Córdoba (15.7%), Granada (15.9%), Jaén (11.4%), and Sevilla (39.4%) participated in the study. Distribution by course was as follows: 35.1% studied second grade of CSE; 16.8%, in third grade of CSE; 23.7%, were fourth graders of CSE; and 24.4%, were in their first year of high school (baccalaureate). Their ages ranged between 12 and 19 years (*M* = 15.05; *SD* = 1.45). These students had a medium socio-economic level, with a 3% dropout rate with 7% foreign pupils in the classroom. The classes were mixed (boys and girls), and it was compulsory for all pupils to take the subject of PE. The distribution according to the teacher’s gender was: 82.2% of students had a male teacher and 17.8% had a female teacher. The work experience in years of the teachers was: *M* = 15.29 years; *SD* = 10.08. Finally, students missing data in any of the seven studied variables were excluded.

### 2.2. Instruments

The Spanish version [13] of the Learning- and Performance-Oriented Physical Education Classes Questionnaire (LAPOPECQ) by Papaioannou [34] was used. The instrument evaluates the perception of the motivational climate generated by the teacher in the PE sessions through 27 items grouped into two factors: task-oriented climate (13 items; e.g., “The PE teacher looks completely satisfied when the students improve after trying hard”) and ego-oriented climate (14 items; e.g., “Students feel very bad when they make mistakes while practicing skills or playing games”). The responses were collected on a Likert-type scale ranging from 0 (totally disagree) to 100 (totally agree) (interval: 0, 10, 20, 30, … 100).

The Spanish version [35] of Papaioannou’s Strategies to Sustain Discipline Scale (SSDS) [23] was used. The instrument consists of 27 items that measure the students’ perception of the strategies used by the teachers to maintain discipline in PE classes. In this study, 24 items grouped into three factors were used: the teacher’s emphasis on intrinsic reasons to maintain discipline (16 items; e.g., “They attract our attention because they make us believe it is important to do well in physical education lessons”); the teacher’s emphasis on introjected reasons to maintain discipline (5 items; e.g., “They make us feel uncomfortable when we are not disciplined”); and the teacher’s indifference to maintaining discipline (3 items; e.g., “They really do nothing to maintain discipline”), with three items being removed from the original scale [35]. The responses were collected on a Likert-type scale ranging from 0 (totally disagree) to 100 (totally agree) (interval: 0, 10, 20, 30, … 100).

The Spanish version [6] of the Sport Satisfaction Instrument with Physical Education (SSI-PE) [36,37] was used. The instrument consists of eight items grouped into two dimensions that measure the satisfaction/enjoyment with PE classes (e.g., “I usually enjoy learning in physical education lessons”) and boredom with PE classes (3 items; e.g., when doing sport, I am usually bored”). The responses were collected on a five-point Likert-type scale ranging from 1 (strongly disagree) to 5 (strongly agree).

### 2.3. Procedure

After obtaining authorization to carry out the study from the management of the educational centers, the students were informed of the purpose and importance of the research and their rights as participants in it, the way to complete each scale, the anonymity of the answers, that these would not affect any qualification in any way, and that they could cease to participate in the study at any time. Data collection was performed by a researcher during the PE lesson. All included participants gave their informed consent prior to participating. The study was carried out following the Declaration of Helsinki and the protocol was approved by the Ethics Committee (Ref: 19002018).

### 2.4. Data Analysis

As a preliminary step, the descriptive statistics, and correlations between variables, as well as the McDonald’s omega coefficient for each factor, were calculated with SPSS v.28, taking values >0.70 to indicate good reliability [38]. In the main analyses, a two-step structural equation model (SEM) was calculated with AMOS v.26 [39] to evaluate the predictive relationships between the motivational climate perceived in PE class and the satisfaction/enjoyment or boredom with this subject, analyzing the mediating role of the different strategies used by the teacher to maintain classroom discipline. In the first step (the measurement model), the robustness of the bidirectional relationships between the model dimensions was evaluated. In the second step, the prediction effects between the variables were analyzed, controlling for the effects of the teacher’s sex and time of service. In this regard, it should be noted that in the event of a violation of the multivariate normality assumption (Mardia’s coefficient = 67.32; *p* < 0.001), the analysis was performed with the maximum likelihood method and the 5000-iteration bootstrapping procedure [39]. The model’s goodness of fit was evaluated considering the following goodness-of-fit indices: chi-squared ratio values and degrees of freedom (χ^2^/df), CFI (comparative fit index), TLI (Tucker–Lewis index), RMSEA (root-mean-square error of approximation) with its 90% (CI) confidence interval, and SRMR (standardized root-mean-square residual). According to authors such as Hu and Bentler [40] or Marsh et al. [41], for the χ^2^/df ratio, values < 5.0 are considered acceptable, as are CFI and TLI values > 0.90, and RMSEA and SRMR values < 0.08. The direct and indirect effects were analyzed taking into account the proposal of Shrout and Bolger [42]: the indirect effects (i.e., mediated) and their 95% CI were estimated with the bootstrapping technique and the significant indirect effect (*p* < 0.05) was considered if their 95% CI did not include the zero value. Although AMOS provides this information after the analyses have been performed, to facilitate interpretation for the reader we include an explanation of how the total effects were calculated. The total effect between two variables (A and C) with mediating role of other variables (e.g., B) is calculated: the sum of indirect effects + direct effect. Finally, in order to provide better interpretation of the results, the total explained variance (R^2^) was considered as a measure of the effect size (ES) [43]. In accordance with Cohen [44], the following ES values were considered: small (<0.02), medium (close to 0.13), and large (>0.26). The CIs (95%) were also calculated to confirm that no R^2^ value was <0.02, as this is the minimum value required for the interpretation.

## 3. Results

### 3.1. Preliminary Results

Table 1 details the descriptive statistics and correlations between the variables (further, a table with the descriptive statistics of each item can be found in Appendix A). With regard to the perception of a motivational climate, it should be noted that the average values for the mastery climate are higher than those for the performance climate. Regarding strategies to maintain discipline, the highest average score corresponds to intrinsic reasons, followed by introjected reasons and indifference. Finally, satisfaction with PE has an average value double that of the average achieved for boredom. Regarding the correlations, the mastery climate presents the highest positive correlations with the performance climate, with intrinsic reasons and with the satisfaction with PE, while the correlations with the introjected reasons, indifference and boredom are negative. Likewise, the performance climate positively correlates with the three discipline variables and satisfaction with PE. Intrinsic reasons have the highest positive relationship with TC and with satisfaction as well as the most notable negative relationship with boredom. Finally, it should be noted that the highest correlation for the introjected reasons is with indifference to maintain discipline, while the correlation between satisfaction and boredom is negative. Regarding reliability, all dimensions achieve values >0.70.

### 3.2. Main Results

In step 1, the model presented acceptable goodness-of-fit indices: χ^2^/df = 2.624, *p* < 0.001; CFI = 0.97; TLI = 0.96; RMSEA = 0.035 (90% CI = 0.031; 0.039), SRMR = 0.035. In step 2 the predictive SEM model also showed an acceptable fit: χ^2^/df = 3.523, *p* < 0.001; CFI = 0.97; TLI = 0.96; RMSEA = 0.045 (90% CI = 0.041; 0.048), SRMR = 0.048. The SEM achieved an explained variance of 32% for indifference to maintain discipline, 28% for the emphasis on intrinsic reasons to maintain discipline, 24% for introjected reasons, 21% for satisfaction/enjoyment with PE, and 17% for boredom with PE. After controlling for the teacher’s sex and service time, in the SEM (Figure 2) one can observe that the TC has a direct positive predictive effect on satisfaction with PE, as well as on intrinsic reasons to maintain discipline, whereas the predictive relationship is negative with introjected reasons and indifference to maintain discipline. It should also be noted that the intrinsic reasons variable acts as a mediator between TC and satisfaction with PE, increasing the total positive effect between the two variables (0.36). On the other hand, the EC has no significant direct relationship with the consequence variables (i.e., satisfaction/enjoyment with PE or boredom with PE), but it should be noted that the teacher’s indifference to maintaining classroom discipline acts as a mediating variable between the perception of EC and boredom. That is, when the climate perceived by students is towards performance and the teacher is indifferent to maintaining discipline, the probability of boredom with the PE classroom increases (0.13) (see Table 2).

## 4. Discussion

The objective of this research was to analyze the mediating role of the teacher’s strategies to maintain discipline between the perceived classroom motivational climate and satisfaction with PE classes. The main results show that TC has an indirect and positive effect on satisfaction with PE mediated by intrinsic reasons to maintain discipline. In addition, a direct and positive effect was found between EC and boredom with PE classes mediated by an indifference to maintaining discipline.

In relation to the hypothesized model, the results reveal that TC has a direct positive effect on satisfaction with PE (H1), as well as a direct positive effect mediated by intrinsic reasons (H4) and a direct negative effect mediated by the indifference to maintaining discipline (H5). Although these relationships are statistically significant, it should be noted that the predictive relationship is higher in the relationship between TC and satisfaction with PE mediated by intrinsic reasons to maintain discipline (H4). These results corroborate previous studies that show the influence of TC on more self-determined strategies towards discipline [23,45]. Accordingly, teachers who orient their methodological strategies towards the process will generate a perception of using intrinsic reasons to maintain discipline, thus leading to greater satisfaction with PE. This could be because TC is a positive predictor of self-determined behaviors, capable of regulating behaviors [46] directed towards involvement in PE classes [47]. In fact, a mastery-oriented classroom climate can mobilize the student’s internal resources, thus developing the self-realization of their objectives and a greater satisfaction and interest in learning [13,48]. Specifically, TC influences the intrinsic reasons to maintain discipline in the classroom, showing itself to be a predictor of satisfactory consequences towards PE. Therefore, establishing motivational climates oriented towards mastery leads to a self-regulated approach regarding reasons for maintaining discipline and having enjoyable experiences in PE.

Continuing with the hypothesized model, the results reveal that EC has an indirect positive effect on boredom with PE classes, mediated by an indifference to maintaining discipline (H5). Likewise, the present study indicates the absence of a direct relationship between EC and boredom, as well as a negative mediation by the intrinsic reasons to maintain discipline, thus rejecting H2 and H6. The perception of an EC generates a positive predictive relationship to an indifference to maintain discipline, fomenting boredom with PE classes (H5). These strategies can lead to students perceiving the teacher as being unconcerned with control and discipline in the classroom, and to them not understanding why they should behave properly, not internalizing these reasons to be disciplined. Our results support other results found in previous research, such as those of Granero-Gallegos et al. [15] and Papaioannou [23], by evidencing an indirect positive relationship between EC and boredom with PE classes. This result could be due to the fact that an indifference to maintaining discipline in the classroom has an influence on subsequent behaviors [35,49]. Specifically, indifference is related to the lowest level of volition and represents the total absence of intention with respect to the preceding behavior [50]. Therefore, it is important to avoid performance-oriented classroom motivational climates so as not to develop boredom in PE classes.

### 4.1. Limitations and Future Prospects

In line with the above results, the SEM helps to understand how characteristic elements of the reasons for applying discipline through a given motivational climate might influence satisfaction or boredom with PE. Another strength of this study is the large sample size used. Despite the above findings, the present research also has certain limitations. First, the convenience and non-randomized sampling method used makes it difficult to interpret the results or to extrapolate them to the whole educational community. Secondly, the information regarding the climates was collected via questionnaires that were self-reported by the students, which can affect the subjective responses about the climates generated in the PE classroom. Future lines of research should study perceived climates using different instruments and consider the perception of the teachers in order to triangulate the data. Third, factors such as didactic knowledge, teacher’s curriculum or teacher’s age may affect the teaching approach in the classroom. In this respect, future research should examine the influence of these factors on students’ perceptions of discipline. Fourth, the age range of the students was between 12 and 19 years. During this age range, there are differences in mental strength and intelligence among the students [51], which may affect the answers to the questionnaires. Finally, authors such as Warburton [52] have argued that students’ perceptions of learning- and performance-oriented climates vary over time, making it difficult to generalize the results. Future research should establish longitudinal designs that measure the influence of climates on satisfaction with the subject over an entire school year, considering the role of the different reasons for maintaining discipline.

### 4.2. Practical Implications

The results of this research highlight the practical importance [53] of creating motivational climates towards learning [54] that influence satisfaction with PE [55]. In this regard, it is necessary for the teacher to establish strategies aimed at generating orientations focused on effort, learning, and personal improvement [56]; for example, using formative assessments (i.e., including positive commentaries through feedback on student progress) [57], giving authority to students (i.e., letting them participate in the instructional process), encouraging recognition (i.e., publicly acknowledging effort), promoting groupings (i.e., creating heterogeneous and cooperative groups in which students collaborate with others to improve their skills) [58], providing temporal flexibility (i.e., in the delivery of learning tasks and activities), and establishing choices in learning tasks and content (i.e., the level of difficulty) [59,60]. Likewise, these findings propose the use of intrinsic reasons to maintain discipline through strategies such as explaining to students the rules for which they must take responsibility, providing autonomy in practice and showing concern regarding the acceptance of responsibility. Similarly, these results suggest that the teacher should avoid generating performance-oriented climates by minimizing the use of coercive language, reducing strategies based on ego involvement and reducing comparative criteria that encourage punishment when doing something wrong and rewards when doing something right.

## 5. Conclusions

The findings of the present study show that the intrinsic reasons to maintain discipline provided by a TC promote satisfaction with PE. On the other hand, they also show the positive influence of an EC on boredom with PE, as measured by indifference towards maintaining discipline. Accordingly, more strategies oriented towards learning and mastery should be applied to encourage satisfaction with the subject and promote and maintain the quality of learning in PE classes while avoiding ego- or performance-oriented climates.

## Figures and Tables

**Figure 1 behavsci-13-00178-f001:**
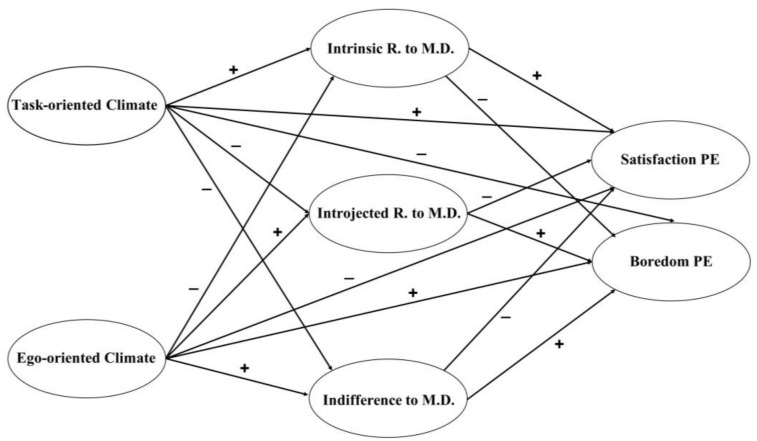
Hypothetical model with the expected relationships. R. = reasons; M.D. = maintain discipline; PE = physical education.

**Figure 2 behavsci-13-00178-f002:**
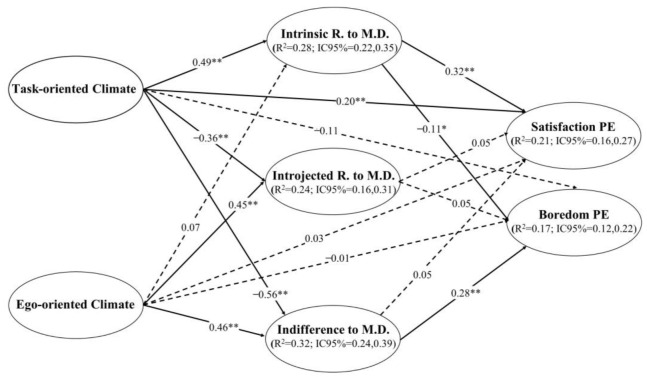
Predictive relationships of the perceived classroom motivational climate on satisfaction/enjoyment and boredom in physical education classes through the mediation of teacher strategies to maintain discipline. Note: ** *p* < 0.01; * *p* < 0.05; PE = physical education; intrinsic R. to M.D. = intrinsic reasons to maintain discipline; introjected R. to M.D. = introjected reasons to maintain discipline; indifference to M.D. = indifference to maintain discipline; R^2^ = explained variance; CI = confidence interval. Dashed lines represent non-significant relationships.

**Table 1 behavsci-13-00178-t001:** Descriptive statistics and correlation between the variables.

Variable	1	2	3	4	5	6	7
1. Task-Oriented Climate		0.45 **	0.49 **	−0.07 *	−0.17 **	0.42 **	−0.26 **
2.Ego-Oriented Climate			0.28 **	0.23 **	0.21 **	0.23 **	0.02
3. Intrinsic Reasons to Maintain Discipline				−0.06 *	−0.12 **	0.40 **	−0.19 **
4. Introjected Reasons to Maintain Discipline					0.50 **	0.00	0.19 **
5. Indifference to Maintain Discipline						−0.03	0.40 **
6. Satisfaction with PE							−0.55 **
7. Boredom with PE							
Range	0–100	0–100	0–100	0–100	0–100	1–5	1–5
Omega	0.81	0.80	0.87	0.82	0.71	0.92	0.84
Mean	68.23	54.45	73.12	51.67	44.75	4.14	2.22
Standard Deviation	17.34	16.65	19.43	21.77	18.66	0.94	0.98
Skewness	−0.21	−0.01	−0.67	0.28	0.67	−1.10	0.94
Kurtosis	−0.55	−0.06	0.05	−0.72	−0.07	1.11	0.18

Note. ** The correlation is significant at the 0.01 level; * the correlation is significant at the 0.05 level.

**Table 2 behavsci-13-00178-t002:** Estimation of the significant standardized parameters and statistics of the mediation model.

Independent Variable	Dependent Variable	Mediator	β	SE	95% CI	
					Inf	Sup
Direct effects						
Task-Oriented Climate	Satisfaction with PE		0.20 **	0.04	0.15	0.26
Task-Oriented Climate	Intrinsic Reasons to Maintain Discipline		0.49 **	0.05	0.38	0.55
Task-Oriented Climate	Introjected Reasons to Maintain Discipline		−0.37 **	0.07	0.32	0.44
Task-Oriented Climate	Indifference to Maintain Discipline		−0.56 *	0.06	0.48	0.65
Ego-Oriented Climate	Introjected Reasons to Maintain Discipline		0.45 *	0.03	0.41	0.54
Ego-Oriented Climate	Indifference to Maintain Discipline		0.46 *	0.05	0.42	0.55
Intrinsic Reasons to Maintain Discipline	Satisfaction with PE		0.32 **	0.05	0.26	0.35
Intrinsic Reasons to Maintain Discipline	Boredom with PE		−0.11 *	0.03	0.04	0.16
Indifference to Maintain Discipline	Boredom with PE		0.28 **	0.04	0.25	0.36
Indirect Effects						
Task-Oriented Climate	Satisfaction with PE	Intrinsic Reasons to Maintain Discipline	0.16 *	0.06	0.07	0.22
Task-Oriented Climate	Boredom with PE	Indifference to Maintain Discipline	−0.16 *	0.05	0.09	0.26
Ego-Oriented Climate	Boredom with PE	Indifference to Maintain Discipline	0.13	0.04	0.07	0.24
Total Effects						
Task-Oriented Climate	Satisfaction with PE	Intrinsic Reasons to Maintain Discipline	0.36 *	0.05	0.28	0.45

Note. β = estimation of standardized parameters; SE = standard error; 95% CI = 95% confidence interval; Inf = 95% CI lower limit; Sup = 95% CI upper limit; PE = physical education; ** *p* < 0.01; * *p* < 0.05.

## Data Availability

The data presented in this study are available on request from the corresponding author. The data are not publicly available due to privacy.

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
