# Peer review of "The Effect of the Motivational Climate on Satisfaction with Physical Education in Secondary School Education: Mediation of Teacher Strategies in Maintaining Discipline"

_behavsci, 2023, doi:10.3390/bs13020178_

Round 1

Reviewer 1 Report

Behavioral Sciences

Manuscript ID: behavsci-2176187

Thank you for the opportunity to review this interesting article entitled: Effect of the motivational climate on satisfaction with Physical Education in secondary school education: mediation of teacher strategies in maintaining discipline.

The manuscript is of great interest and is well structured and theoretically justified. The objective is pertinent and the hypotheses are justified. Likewise, the methodology used to answer the objective is adequate and very well explained. The sample is very large and the practical implications proposed can also be highlighted.

This article deals with a fundamental issue in the teaching-learning process such as discipline. It reminds me of readings on classroom environment and the psychosocial relationships established between students and teachers, which Barry Fraser from Curtin University in Australia dealt with back in 1989 (Classroom environment instruments: Development, validity and applications). In my case, it was useful for my doctoral thesis. So thank you very much for allowing me to read this article.

In this reading, the aim of this study was to analyse the mediating role of strategies for maintaining classroom discipline between the motivational climate generated by the teacher and students' satisfaction with physical education classes.  The research design was very well structured and clearly presented. A relevant and adequate sample of 2147 secondary school PE students (1050 males; 1097 females) participated. A structural equation model was estimated with the latent variables controlled for gender and teacher length of service and using the scales of motivational climate, teacher strategies for maintaining discipline in class and student satisfaction with PE. The results of the model highlight the importance of intrinsic strategies in maintaining discipline, which act as mediators between the motivational climate towards learning and students' satisfaction with Physical Education.

I highlight and agree with the results of this study, as they reveal the influence of a performance-oriented climate in the prediction of boredom in class when the teacher shows indifference towards the maintenance of discipline. Therefore, as teachers we cannot be indifferent towards maintaining discipline.

On a formal level, it is very well structured, with clear and concise language. It uses current bibliographical references for the subject matter.

However, I would ask you to correct the following:

In the "design and participants" section, the proportion of the sample from each of the provinces could be specified, as well as the courses to which the participants belong. Likewise, the authors could specify the criteria for inclusion of the participants in the study. Also, given that the SEM has controlled for gender and length of service of the teachers, this data could be specified (proportion of gender and length of service as a teacher).

Please review heading 4.1 and write it in English.

In the References section, those that are in Spanish, should be followed by their English translation in square brackets [English translation]. For example, references: 1, 35, 38 and 43.

Best regards and congratulations.

Author Response

We thank the reviewers for his/her constructive comments and his/her thorough revision of the manuscript. Below we answer his/her questions and concerns, including explicitly the changes made in the manuscript as well.

Comment: In the "design and participants" section, the proportion of the sample from each of the provinces could be specified, as well as the courses to which the participants belong.

  • Response: In the "Design and participants" section, the proportion of the sample from each of the provinces (Almería, 17.6%; Córdoba, 15.7%; Granada, 15.9%; Jaén, 11.4%; Sevilla, 39.4%) could be specified, as well as the courses (“Distribution by course was as follows: 35.1% studied second grade of Compulsory Secondary Education(CSE); 16.8%, in third grade of CSE; 23.7%, were fourth graders of CSE; y 24.4%, their first year of high school (bachillerato)” to which the participants belong.

Comment: Likewise, the authors could specify the criteria for the inclusion of the participants in the study.

  • Response: Suggestions made by the reviewer were made in the "Design and participants" section. Inclusion criteria have been added: “The following inclusion criteria were established to participate in this study: (i) to be a student of the second, third, or fourth grade of CSE, or first year of high school (bachillerato) of a public secondary school from the autonomous region of Andalusia; ii) to provide informed consent for participation; iii) regular attendance at classes; iv) fill out the questionnaire completely.”

Comment: Also, given that the SEM has controlled for gender and length of service of the teachers, this data could be specified (proportion of gender and length of service as a teacher).

  • Response: Suggestions made by the reviewer were made in the "Design and participants" section

Comment: Please review heading 4.1 and write it in English.

  • Response: Suggestions made by the reviewer were made.

Comment: In the References section, those that are in Spanish, should be followed by their English translation in square brackets [English translation]. For example, references: 1, 35, 38 and 43.

  • Response: Suggestions made by the reviewer were made.

Reviewer 2 Report

Authors examined whether and how much teachers’ classroom management strategies mediate the association between task- and ego-oriented climate and PE class satisfaction, using a large sample and rigorous SEM mediation analysis.

SEM procedures are clearly reported, including the measurement model and structural model with descriptions of estimation process such as bootstrapping. However, I propose a few methodological suggestions for revision. First, it is not clear how authors handled missing data if any. Authors also never introduced mastery and performance climate before the results section, so it would be helpful if they briefly define the concepts or change the labels if they are equivalent to TC and EC. Authors report they used Likert-type scales ranging from 0 to 100. It would be helpful what intervals within responses look like (line 140). For instance, respondents could choose from 1, 2, 3…100 or the interval was every 10, such as 10, 20, 30?

On p. 6, I encourage the authors to provide a table or formula that illustrate how the total effect was computed. For example, readers who are not familiar with mediation analysis may be lost when seeing statistics such as .36 in line 228 without any explanations on how the number resulted in. Without supporting evidence or report, it is also unclear what exactly “increasing the total positive effect” means in line 227 on p. 6. In lines 255-257, authors report “the predictive relationship is higher in the relationship between TC and satisfaction” but it is not clear higher “than what.”

There are also a few English grammar issues (e.g., the 1st sentence in introduction, 1st sentence of section 2.2.) and some statements on p. 3 do not mesh with signs on Figure 1 on p. 3.

Author Response

We thank the reviewers for his/her constructive comments and his/her thorough revision of the manuscript. Below we answer his/her questions and concerns, including explicitly the changes made in the manuscript as well.

Authors examined whether and how much teachers’ classroom management strategies mediate the association between task- and ego-oriented climate and PE class satisfaction, using a large sample and rigorous SEM mediation analysis. SEM procedures are clearly reported, including the measurement model and structural model with descriptions of estimation process such as bootstrapping. However, I propose a few methodological suggestions for revision.

Comment: First, it is not clear how authors handled missing data if any.

  • Response: This clarification has been added at the end of the paragraph “design and participants”.

Comment: Authors also never introduced mastery and performance climate before the results section, so it would be helpful if they briefly define the concepts or change the labels if they are equivalent to TC and EC.

  • Response: The following clarification has been added in the introduction (lines: 63-68, revised manuscript) “TC refers to that centred on task-focused cooperation between learners, occurs when the situational characteristics of the teacher or classmates are mindful of supporting effort, and success is focused on both the learning process and on intrapersonal criteria related to effort and personal improvement. In contrast, EC is defined by mastery of the task at any cost, where success is based on normative and interpersonal criteria, with a punitive response to errors and rivalry among students”.

Comment: Authors report they used Likert-type scales ranging from 0 to 100. It would be helpful what intervals within responses look like (line 140). For instance, respondents could choose from 1, 2, 3…100 or the interval was every 10, such as 10, 20, 30?

  • Response: Thank you very much for your comment. This clarification about Lickert-type scales ranging from 0 to 100 has been added in the paragraph “Instruments”, both in The Learning and Performance Oriented Physical Education Classes Questionnaire (LAPOPECQ) as Strategies to Sustain Discipline Scale (SSDS). In the lines 153 and 164-65 (revised manuscript) you can find this clarification: (interval 0, 10, 20, 30, … 100).

Comment: On p. 6, I encourage the authors to provide a table or formula that illustrate how the total effect was computed. For example, readers who are not familiar with mediation analysis may be lost when seeing statistics such as .36 in line 228 without any explanations on how the number resulted in. Without supporting evidence or report, it is also unclear what exactly “increasing the total positive effect” means in line 227 on p. 6. In lines 255-257, authors report “the predictive relationship is higher in the relationship between TC and satisfaction” but it is not clear higher “than what.”

  • Response: Thank you very much for your comment. The next information has ven included in the paragraph “Data análisis”. Although AMOS provides this information after the analyses have been performed, to facilitate interpretation for the reader we include the explanation of how the total effects were calculated. The total effect between two variables (A, C) with mediating role of other variables (e.g. B) is calculated: sum of indirect effects + direct effect” (Lines: 204-208, revised manuscript).
  • For example: indirect effects [(direct effect of A (TC) on B (intrinsic reasons) x (direct effect of B (intrinsic reasons) on C (satisfaction PE)] + direct effect of A (TC) on C (satisfactin PE). That it to say: [(TC on intrinsic reasons PE; i.e., .49) x (intrinsic reasons on satisfaction PE; i.e., .32)] + TC on satisfaction PE (i.e., .20) = .36 (.16 + .20 = .36)

Comment: There are also a few English grammar issues (e.g., the 1st sentence in introduction, 1st sentence of section 2.2.) and some statements on p. 3 do not mesh with signs on Figure 1 on p. 3.

  • Response: Signs on Figure 1 have been revised and modified. As well, English grammar has been revised by a native profesional translator (1st sentence in introduction) and 1st sentence on section 2.2 (however, this is original title of the scale)

Reviewer 3 Report

Dear Authors

congratualtion for the work. 

I suggest some minor revision:

- Previous Works -- change in Previous surveys

- 4.1 Futuras --> probably is a refuse change in further

please, specify the area of granada, sevilla...are in the same region of Spain? very close? are all the school in urban or rural context? can you find different traditional, environment and cultural approach to the school in your selected city?

please, specify in limitation that any control on the CV of PE teacher are made. infact, the educational background, the age of teacher, the age of work could be affect the didactical approach.

- specify in limitation the large range of age (12-19) because the intelligence, the approach, the mental strength in children-adolescent-adults are differents (please use references because the relationship between students and teachers become different)

- in results section could be important provide a simple general data about mean and sd of the results in questionnaire (istogramm by items)

best

Author Response

We thank the reviewers for his/her constructive comments and his/her thorough revision of the manuscript. Below we answer his/her questions and concerns, including explicitly the changes made in the manuscript as well.

Dear Authors

congratualtion for the work. 

I suggest some minor revision:

Comment:- Previous Works -- change in Previous surveys

  • Response: Suggestions made by the reviewer were made (line: 46, revised manuscript).

Comment:- 4.1 Futuras --> probably is a refuse change in further

  • Response: Suggestions made by the reviewer were made.

Comment: please, specify the area of granada, sevilla...are in the same region of Spain? very close? are all the school in urban or rural context? can you find different traditional, environment and cultural approach to the school in your selected city?

  • Response: Suggestions made by the reviewer were made and this aclarations have been specified in the paragraph “Design and participants”.

Comment: please, specify in limitation that any control on the CV of PE teacher are made. infact, the educational background, the age of teacher, the age of work could be affect the didactical approach.

  • Response: This specification has been added in limitations (lines: 317-317, revised manuscript): “Third, factors such as didactic knowledge, teacher's curriculum or teacher's age may affect the teaching approach in the classroom. In this respect, future research should examine the influence of these factors on students' perceptions of discipline.”

Comment: - specify in limitation the large range of age (12-19) because the intelligence, the approach, the mental strength in children-adolescent-adults are differents (please use references because the relationship between students and teachers become different).

  • Response: This specification has been added in limitations (lines: 318-320, revised manuscript): Fourth, the age range of the students was between 12-19 years. During this age range, there are differences in mental strength and intelligence among the students [51], which may condition the answers to the questionnaires.
  • Glück, J.; Scherpf, A. Intelligence and wisdom: Age-related differences and nonlinear relationships.  Aging2022 37 649–666. https://doi.org/10.1037/pag0000692

Comment:- in results section could be important provide a simple general data about mean and sd of the results in questionnaire (istogramm by items)

  • Response: In the results section (line 216, revised manuscript) there is a reference to a table that can be found in Appendix A (able with descriptive statistics has been added in Appendix A).

Round 2

Reviewer 2 Report

Thank you for effectively addressing the comments.